

# Hybrid computational and real data-based positioning of small cells in 5G networks

Flávio Henry Ferreira[1], Fabrício José Brito Barros[1], Miércio Cardoso de Alcântara Neto[1], Evelin Cardoso[2], Carlos Renato Lisboa Francês[1] and Jasmine Araújo[1]

[1] Post Graduate Program in Electrical Engineering, Institute of Technology of Federal University of Pará, Federal University of Pará, Belém, PA, Brasil
[2] Computer Systems Department, Federal Rural University of the Amazon, Capitão Poço, Pará, Brasil

## ABSTRACT

One of the key technologies in smart cities is the use of next generation networks such as 5G networks. Mainly because this new mobile technology offers massive connections in densely populated areas in smart cities, thus playing a crucial role for numerous subscribers anytime and anywhere. Indeed, all the most important infrastructure to promote a connected world is being related to next generation networks. Specifically, the small cells transmitters is one of the 5G technologies more relevant to provide more connections and to attend the high demand in smart cities. In this article, a smart small cell positioning is proposed in the context of a smart city. The work proposal aims to do this through the development of a hybrid clustering algorithm with meta-heuristic optimizations to serve users, with real data, of a region satisfying coverage criteria. Furthermore, the problem to be solved will be the best location of the small cells, with the minimization of attenuation between the base stations and its users. The possibilities of using multi-objective optimization algorithms based on bioinspired computing, such as Flower Pollination and Cuckoo Search, will be verified. It will also be analyzed by simulation which power values would allow the continuity of the service with emphasis on three 5G spectrums used around the world: 700 MHz, 2.3 GHz and 3.5 GHz.

## INTRODUCTION

The emerging concept of Smart cities is being promoted with the deployment of 5G technologies. Considering the increasing amount of industrialization with urbanization, the huge demand for resources and their ubiquitous use are catalyzing the emergence deployment of smart city technologies and applications. All urbanization enablers like transportation and mobility, health care, natural resources, electricity and energy, homes and buildings, commerce and retail, society and workplace, industry, agriculture and the environment, are hugely dependent of a suitable communication and safe capabilities to support these smart cities application domains.

Meanwhile, the fifth generation of wireless communications, the 5G system, is currently being integrated, with an extensive range of applications and frequency channels for its operation. According to *Dahlman et al. (2014)*, 5G operation aims at a 1,000 times greater

Corresponding authors
Flávio Henry Ferreira,
henryferreira014@gmail.com
Jasmine Araújo, jasmine@ufpa.br

traffic capacity and a pulled bandwidth capacity capable of working with a latency response of 1 ms with data rates in the order of 1 to 10 Gbps. The development of 5G systems is divided into indoor and outdoor spreads. Generally, the sub-6 GHz band is applied outdoors because it is easier to transmit and propagate, and communication companies are already testing and applying 5G systems in this band for commercial purposes, in different parts of the world, as reports from *Viavi (2022)* have shown. In Brazil, it is no different—all major operators already have 5G systems in operation for several capitals in the country. *AgênciaBrasil (2021)* and *Reuters (2021)* inform that the main frequencies to operate 5G in the country, according to the government body Anatel, are 700 MHz, 2.3 GHz and 3.5 GHz for outdoor systems, and 26 GHz for indoor deployments.

*GSMA (2021)* states the band allocation of auctioned 5G spectrums in Brazil. The sub-6 GHz bandwidths are 20 MHz for 700 MHz (10 MHz per operator), 90 MHz for 2.3 GHz (40 or 50 MHz per operator), and 400 MHz for 3.5 GHz (80 to 100 MHz per operator). The 3.5 GHz spectrum is, currently, the most used for 5G applications around the world, and it is the range that possesses the greater number of proposed devices in the literature. Examples can be found in *Li et al. (2021)* and *Kapoor, Mishra & Kumar (2021)*.

Along the difficulties tied to the deployment of 5G heterogeneous networks (5G HetNet), are the challenges to optimize their user coverage and user capacity, allowing for an increased number of services provided whilst keeping network costs low. And, as *Rappaport et al. (2013)* has stated, 5G coverage should be available everywhere, to anyone. That is, user coverage needs to be as closer to a hundred percent as ever.

A number of studies and surveys have dealt with the necessity of coverage optimization. *Agiwal et al. (2021)*, for instance, dedicates a whole survey on the applications of 4G–5G inter-operations, and how those can be better achieved. This is because implementation of 5G networks till this day are much costing, and changes cannot be applied overnight. Meanwhile, 4G-LTE cells can provide service coverage while 5G is still expanding. Another survey that is worth noticing, written by *Shayea et al. (2020)*, focuses on user mobility management and how user equipments (UEs) are prone to disconnect if there are no satisfactory solutions to coverage, capacity and handoff problems and challenges.

The study herein proposed aims to provide a solution to the coverage problem for future 5G network applications, focusing especially in the range of small cells. The positioning of small cells is a key concept of densification offering a potential solution for the ultra-dense traffic in Smart cities. Otherwise, to add to the traditional cell planning in this work two types of computational intelligence techniques will be tested, *i.e.*, metaheuristic optimization through the utilization of a bioinspired computing algorithm (BIC) and a clustering technique. It is possible to group a set of users into an intelligent network coverage system, that aims to not only optimize the number of small cells but also deal with energy efficiency measures (such as controlling the transmitted power used in the cells).

Bioinspired computational methods are mainly based on natural selection. They are set to mimic the natural behavior of nature, in which the best and most surviving individuals prevail. With that in mind, these bioinspired algorithms serve as good optimization methods for mathematical and engineering problems, especially those where metaheuristic techniques (trial and error) can be applied to achieve one or more concrete goals. They

have been applied to a multitude of areas where non-linear, multimodal optimization is required. *E.g.*, *Li et al. (2022)* cite some areas of robotics where they might be useful, *Nguyen et al. (2020)* exposes some challenges in smart energy management that can be overcome with bioinspired solutions, and *Gill & Buyya (2019)* shows that some of them are even used on big data analysis and as aid to digitization of important documents into digital libraries.

The clustering method chosen for this application is K-means, which is extensively utilized in the literature for its simplicity and efficiency *Ahmed, Seraj & Islam (2020)*. As for the bioinspired methods, two are to be tested in conjunction with K-means: the Cuckoo Search (CS) and the Flower Pollination Algorithm (FPA). As the aim of the work is to both maximize user coverage and minimize transmitted power, it is needed to use their multi-objective counterparts (that is, MOCS and MOFPA).

By using real data from open and free database OpenCellid, see *Khan, García-Armada & Escudero-Garzás (2020)*, it is possible to pre-select cells with the greatest amount of user traffic in 4G-LTE in order to plan out how future 5G small cells shall behave in order to provide good coverage of service to users. More details about this are to be explained in the Methodology section.

The proposed hybridization of clustering and bioinspired algorithms is to be tested in simulations to determine an optimal user coverage for the three aforementioned frequencies that have been auctioned to operate in Brazil: 700 MHz, 2.3 GHz and 3.5 GHz. In total, two hybrid algorithms have been produced: MOCS + K-means (MOCS-KM) and MOFPA + K-means (MOFPA-KM).

The main contributions of this study are the provision of a metaheuristic method to achieve optimal network coverage with low-power small cells, and to provide data that can be adapted to an densely urban but with a rainforest climate such as the city of Belém, Brazil—which is where our OpenCellid data is from. A considerable area of the city has been selected to test the intelligent UE clustering simulations for 5G small cell implementation.

The article is organized into the following sections: Related Works discourses about some of the state-of-the-art solutions for coverage and capacity optimization for 5G as well as bioinspired/clustering algorithm hybrids; Methodology explains how the study was conducted, the theory behind the algorithms utilized and gives information on the propagation model chosen for path loss and user coverage modeling; the Results demonstrate the simulation of the algorithms for the different frequency ranges; and Conclusions expose our final considerations of the study.

## RELATED WORKS

This section is to be divided into three topics: Coverage and capacity optimizations for applications in 5G systems, Heterogeneous wireless networks and 5G architecture, and Bioinspired algorithm hybridizations focusing on clustering of data.

## Coverage and capacity optimization of 5G networks

An analytical study on the coverage, handoff and cost optimization for heterogeneous 5G networks (5G-HetNet) has been written by *Ouamri et al. (2020)*. A path loss model is suggested for both Line-of-Sight (LoS) and Non-Line-of-Sight (NLoS) situations, with different path loss exponents, and the handling of coverage and handoff probability is done by stochastic geometry with values in SINR (Signal to Interference-Plus-Noise Ratio). By the method of Cellular Network Planning (CNP), network investment cost issues are optimized.

*Khan, García-Armada & Escudero-Garzás (2020)* talks about a heuristic method to predict the employment of 5G base-stations in Spain by obtaining user and traffic data from LTE networks, using the free database of OpenCellid. This dimensioning of a 5G network takes into consideration the various aspects of this technology: heterogeneous architecture, the necessity for a high Quality of Experience (QoE) and smart resource allocation. Their objectives are achieved by separating the highest traffic areas, and then deciding where 5G cells are to be deployed.

*Khan et al. (2022)*, however, improves on their model of 5G planning by assigning a clustering algorithm to the task of deploying 5G base-stations into high data traffic areas. A K-means algorithm has been utilized, using the Elbow heuristic as a benchmark. Not only does it deal with the coverage characteristics of the network, but also demonstrates an extensive study on its capacity dimensioning. The goal of the study is to decrease the network cost, as well as provide a more robust way to interpret the data for network planning decision.

Another article that talks about coverage area optimization for 5G is found in *Wang, Lee & Wu (2020)*. It is mathematically complex, as the goal is to provide numerical solutions for the employment of macro and small cells in 5G. Therefore, different clustering approaches are suggested by the authors, as well as models to prevent noise and interference. Numerical case examples are given for different scenarios and, as it deals with heterogeneous networks, the coverage simulations can be seen with generally one or two macro cells and several small cells around them.

*Jia, Ji & Chen (2019)* denotes another mathematically complex problem: the intra-clustering organization and interference issues of 5G millimeter wave (mmWave) networks. Given that mmWave has very little coverage, the usage of Pico cells and Femto cells proves to be necessary. This amounts to a large number of cells and nodes within the network that cluster up and results to large amounts of inter and intra-cluster interferences. The article then lists some of the differences between Pico and Femto cell user association, and then derive said UE associations and Laplace transforms of interference *via* stochastic geometry.

Meanwhile, *Bektas et al. (2021)* deals with the planning of private 5G networks. In situations which require fast or even temporary network solutions, the authors have provided an unsupervised machine leaning (UML) technique for base-station placement. It takes into consideration the boundaries and environmental variables of the area to be covered, as well as service quality (in this case, a signal greater than −90 dBm). Ray Tracing

simulations on a virtual environment emulating the Monaco Grand Prix are made to validate and test the outputs of the UML implementation.

*Jain, Lopez-Aguilera & Demirkol (2021)*, in turn, proposes a framework for 5G capacity optimization based on User Association and Resource Allocation (AURA-5G). It aims to optimize for both maximum sum of throughputs and best bandwidth allocation within the network. This framework is then applied to be tested in many distinct architectural applications of 4G and 5G, such as Enhanced Mobile Broadband (eMBB) and Massive Machine Type Communication (mMTC). The study demonstrates that this framework improves throughput and other metrics such as latency when compared to baseline user association techniques.

## Heterogeneous wireless networks and 5G architecture

As data volume and demands increase with the higher connection data rates of 5G, it is necessary to implement different ways of handling user access to improve the connectivity and capacity of future networks. Hence, 5G implementations aims to provide heterogeneous access and network slicing to meet different data criteria. Here, we shall discourse on some works on the matter, given that it is a fundamental field of study for future applications.

In *Haile, Mutafungwa & Hämäläinen (2020)*, there are sections dedicated on elucidating about ownership models and heterogeneity of architectures in small cell-based 5G-NR networks. Given that the article focuses on the hyperdensification due to growing user demands, it claims that operator-based small cells may not be sufficient to support user data traffic. So, neutral third-parties (governmental or private companies) or even user-deployed small cells may be employed to deal with such problems.

The authors then discourse that a shift to a "service-oriented" paradigm for network planning will become more common eventually, and that 5G network slicing is one of its main features. Network slicing is a way to utilize distinct, end-to-end logical networks taking into consideration different types of user demands whilst still operating within the same physical structures—see *Zhang (2019)*.

One common architecture and user access standard for 5G addressed in many works such as *Harper & Sirotkin (2020)*, *Haile, Mutafungwa & Hämäläinen (2020)* and *Bertenyi et al. (2018)* is NG-Radio Access Network (NG-RAN). It is classified as a heterogeneous network as it aims to function with nodes that can provide either 5G New Radio (5G-NR) access or 4G-LTE access (ng-eNB). Its operation can be standalone or non-standalone. Standalone means that only the 5G core architecture (5GC) will be utilized, whilst non-standalone is an integration of NR and LTE-EPC architectures allowing user access for both 4G and 5G. Moreover, network virtualization of 5G base-stations can allow traffic separation into central units (CU), distributed units (DU) and radio units (RU), as well as provide cloud compatibility (C-RAN).

Another technique to improve user access is non-orthogonal multiple access (NOMA), which can be applied to a plethora of situations within 5G networks, such as Femto, Pico and Micro Cell 5G coverage and Networked Flying Platforms (NFPs). According to *Lv et al. (2020)* and *Diamanti et al. (2020)*, as opposed to orthogonal multiple access (OMA)

that allocates resources for each user separately, NOMA utilizes multiplexing in the power domain in order to accommodate multiple users into the same time/frequency resources, promoting a better utilization of available bandwidth, *via* a method called superposition coding (SC). The management of interference between users is then processed by secure interference cancellation (SIC), a technique that decodes this multiplexed signal at the receiving end.

Given its extremely focused on power allocation and that channel state information (CSI) can be difficult to acquire on this type of technique, optimizations for NOMA network tend to lie under these subjects. Examples are the aforementioned work of *Diamanti et al. (2020)* that proposes power allocation optimization *via* reinforcement learning (RL) and contract theory (CT), *Xu & Cumanan (2017)* that proposes the same problem but considering the statistical CSI on transmitter and using the α-fairness method, and *Fang et al. (2017)* that considers imperfect CSI and aims to improve resource allocation within a multi-carrier NOMA (MC-NOMA) framework.

*Kuribayashi et al. (2020)* has developed a cell range expansion technique based on a particle swarm algorithm. This work fills both roles of providing coverage planning and expansion while maintaining a HetNet context. The objective is to achieve said small cell coverage expansion by maximizing the number of users whose downlink connection requirements are met. The algorithm then balances the load of small cells *via* maximum SINR of users, and uses it to model an objective function that aims to compute a cell range Expansion bias (CRE). Conventional PSO and other particle-based approaches have been tested against their proposed algorithm, the PSO-Based CRE Bias (PCB), which has shown to be better for this type of application.

## Bioinspired algorithm hybridizations for data clustering

*Jensi & Jiji (2015)* have accomplished to generate a FPA hybrid with K-means, a similar method to the study presented herein. In the article, they present this technique to test and clusterize eight different datasets, with promising results. It is attested that the hybridization has given better average fitness results for all tests, thus proving it to be more efficient.

However, the hybridization process shown in the article takes the current best solution in the FPA and executes a local search around it using the K-means. This is different from the implementation we have provided, to be further explained in the Methodology section.

*Hatamlou (2017)* is another article that deals with nature-inspired algorithms, with their proposed technique of utilizing particle swarm optimization (PSO) with the big bang-big crunch (BB-BC). The objective, like in *Jensi & Jiji (2015)*, is to provide an efficient method of data clustering, and both of them use five equal datasets (Iris, Wine, Glass, CMC, Cancer) in order to test their techniques. However, for all the same datasets, the PSO-BB-BC hybrid has shown better average fitness than its FPA-KM counterpart.

Another robust work in the subject is found in *Logesh et al. (2020)*. Based on TripAdvisor travel recommendations data, the authors propose several hybrid solutions. These are based on swarm intelligence algorithms, such as an improved PSO model, Brainstorm Optimization (BSO), Quantum-Behaved Brainstorm Optimization (QBSO)

and the Immune Genetic (IG) technique. In most cases, the best optimizer is the novel QBSO-IG hybrid that they have proposed themselves. This study also provides many details on recommendation systems and draws a comparison on many other articles that have dealt with its data clustering optimizations.

*Khan, Aftab & Zhang (2019)* have proposed a clustering technique for Flying Ad-Hoc Networks (FANETs), named Bio-Inspired Clustering Scheme for FANETs (BICSF), that consist of organizing drones in the air based on their positioning and energy management. The goal is to reduce energy consumption, elevate the air time of the devices and minimize the time that the drones take to organize themselves into clusters. The hybrid is between the glowworm swarm optimization (GSO) and krill herd (KH) algorithms, and it is tested against more commonly found bioinspired ones—Grey Wolf Optimization (GWO) and Ant Colony Optimization (ACO), to specify.

*Pitchaimanickam & Murugaboopathi (2020)* also proposes a hybrid approach for network clustering, however, with an emphasis on ameliorating battery life and information management in wireless sensor networks. The creation of clusters itself is done by a LEACH-C algorithm, which is then optimized with a conjuction of a PSO and the Firefly Algorithm (FA). HFAPSO, as the hybridization is abbreviated, proves to be considerably more efficient than the separated techniques in a simulation of 100 sensors and an area of $(100 \times 100)$ m$^2$. Results have shown that not only does it better coordinates the clusters, but it sustains the battery life of sensor nodes for longer.

*Cao et al. (2021)* demonstrates a heterogeneous Wireless Sensor Network (WSN) coverage area optimization, in which a Chaos-Improved Social Spider Optimization (CSSO) is used to accomplish the task. The solution aims to deploy the sensors by saving as much energy consumption and network cost as possible, as well as reducing coverage redundancy and trying to cover blind spots as best as possible.

Lastly, a study that also uses K-means in a hybrid bioinspired computing setting is conducted by *Aswani, Kar & Vigneswara Ilavarasan (2018)*. It is combined with a Firefly Algorithm, but also with a modified, Lévy flight and chaos-induced version of it (LFA-chaos). This clustering technique is then tested against a Fuzzy C-Means (FCM) algorithm, and its application is to find spam accounts on social media website Twitter. It uses many significant behavioral factors of the user as input to solve this problem, such as its hashtag frequency, number of mentions, tweet count, follower count and many more, up to a total of 13. The Lévy flight applied to this FA is similar to the one used in the FPA exposed in the Methodology section.

## Contributions

The main contribution to our study is the utilization of a fairly simple low complexity bioinspired computing and clustering hybrid in order to solve a cell planning (CP) problem by means of small cell coverage and transmitted power optimization.

This study also utilizes real-life data of LTE by the methods of *Khan, García-Armada & Escudero-Garzás (2020)* in order to provide a life-like user deployment in the city of Belém, Brazil—with suburban to urban characteristics and within the Amazon rainforest biome. The works of *Khan, García-Armada & Escudero-Garzás (2020)* and *Khan et al. (2022)* may

be evolved in user selection and data mining but still leave a gap to come up with more developed, faster computational methods to solve 5G coverage problems—the former study used a heuristic created by the authors themselves, and the latter used only a clustering algorithm. The hybrid solution present herein is a better and more developed technique overall, therefore it shall produce better coverage optimization results.

Cell planning is considered to be a NP-hard problem, therefore, a solution in real time is unlikely to be found and an exact solution is hard to find in polynomial time, as explained by *Weicker et al. (2003)* and *Taufique et al. (2017)*. Bioinspired algorithms, however, tend to have low algorithmic complexity but are able to solve highly complex problems—see *He et al. (2017)*. They can be useful tools by providing a metaheurisitc approach that takes less time than other techniques to produce satisfactory network planning results—*e.g.*, the studies of *Kaur, Pal & Singh (2020)* on the matter. Thus, another contribution of this work is the Running Time analysis conducted to attest that such NP-hard, non real-life problem can be further optimized in a matter of a dozen or couple dozen of minutes only.

As a secondary objective, the work herein can also become a reference for further coverage and capacity planning of 5G networks in suburban and urban environments of the Amazon rainforest region, or even other regions of Brazil, by the usage of real-life user data estimation and dimensioning—which enhances the verisimilitude of simulational results.

## METHODOLOGY

In this section, details about the objectives, the methods and optimization techniques used in this study are elucidated.

In general terms, the aim of the study presented herein is to acquire real-life LTE user data from OpenCellid to aid in the deployment of 5G cells in a four-neighborhood area from the city of Belém, Brazil. The involved neighborhoods are Sacramenta, Barreiro, Miramar and Pedreira, and the area is represented in Fig. 1.

For that matter, by selecting the LTE cells with the greater traffic according to OpenCellid data, and inspired by the data mining process of *Khan et al. (2022)*, a total of 19 LTE cells were selected to provide the highest data traffic areas and aid our deployment proposal. Therefore, by taking learning data from these cells, we have made a hybrid clustering/bioinspired technique that aims to solve the coverage problem in 5G frequency bands of 700 MHz, 2.3 GHz and 3.5 GHz.

Two problems to be optimized show up when thinking of providing coverage for a medium-to-high density urban area so large such as this: the number of $k$ cells (or clusters of users) that is required to achieve a good user coverage percentage, and also, for matters of energy efficiency, how much power $P_t$ should be applied to these cells in order to reach the best coverage by spending less energy. Given that our approach aims to minimize both these variables, network costs should also be minimized throughout the process. So, a simulation with an $N$ number of users normally distributed in the aforementioned area is drawn to test how much coverage the proposed metaheuristic setup will achieve.

The Methodology from here is separated in various subsections. Firstly, the bioinspired algorithms shall be revised, then an explanation on the K-means clustering algorithm is

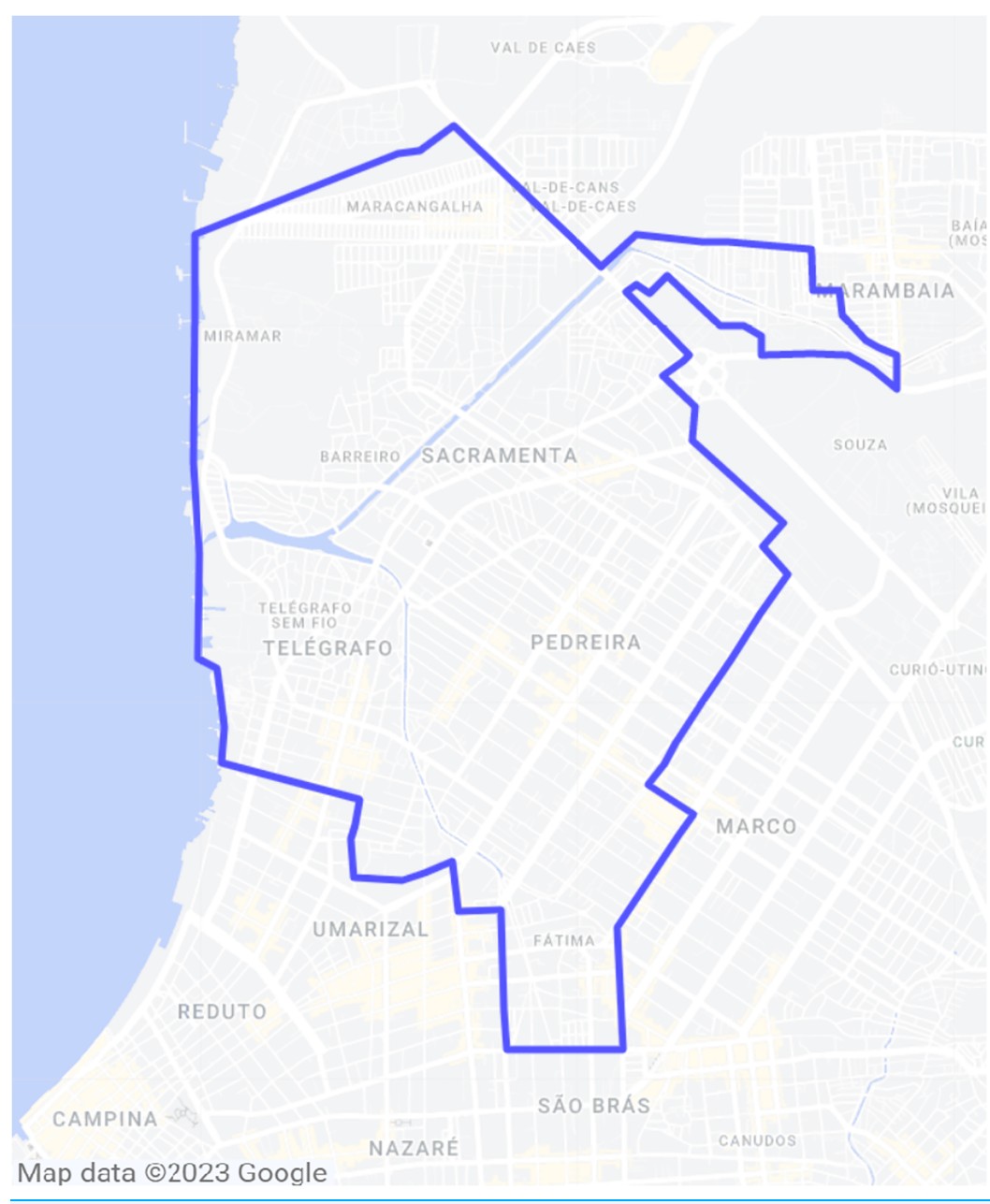

**Figure 1 Area of interest inside Belém, Brazil.**

given. Lastly, the hybridization process is explained, as well as the path loss propagation model for producing the objective functions.

Also, Table 1 denotes all variables, constants and parameters that are present in equations and results throughout the article with brief explanations and, when present, their units.

**Table 1 List of variables, constants and parameters.**

| Symbols | Descriptions |
| --- | --- |
| $A_{bm}$ | Base median path loss (ECC model) |
| $A_{fs}$ | Free-space attenuation (ECC model) |
| $GF_r$ | Receiving antenna gain factor (ECC model) |
| $GF_t$ | Transmitting antenna gain factor (ECC model) |
| $G_r$ | Receiving antenna gain |
| $G_t$ | Transmitting antenna gain |
| $L$ | Pollination strength for global pollination (MOFPA) |
| $L_b$ | Lower bounds of the search space (MOCS and MOFPA) |
| $N_{connected}$ | Number of users connected to the network |
| $N_i$ | Maximum number of iterations (MOCS and MOFPA) |
| $N_{outage}$ | Number of users disconnected from the network |
| $N_{users}$ | Number of total users in the optimization search space |
| $PL_{ECC}$ | Path loss given by the ECC model (in dB) |
| $P_a$ | Nest discard probability (MOCS) |
| $P_{diff}$ | Differential between optimized transmitted power and its minimum value (in dBm) |
| $P_r$, or $R_x$ | Received power (in dBm) |
| $P_t$, or $T_x$ | Transmitted power (in dBm) |
| $U_b$ | Upper bounds of the search space (MOCS and MOFPA) |
| $X_i^t$ | Position of an individual in the search space, for iteration t (MOCS and MOFPA) |
| $X_i^{t+1}$ | Position for next iteration t+1 of an individual in the search space (MOCS and MOFPA) |
| $a_k$ | Set of centroid positions (K-means) |
| $d$ | Haversine distance between a base-station and a user (in meters) |
| $f_{avg}$ | Average fitness of the best generation produced by the optimization |
| $f_{best}$ | Best fitness produced by the optimization, overall |
| $f_c$ | Central frequency (in MHz or GHz, as specified) |
| $f_i$ | Single objective i, in a multi-objective function |
| $f_{max}$ | Maximum fitness found within the best generation of the optimization |
| $g*$ | Best current solution (MOFPA) |
| $h_{BS}$ | Base-station antenna height (in meters) |
| $h_{UE}$ | User antenna height (in meters) |
| $k$ | Number of clusters given by the optimization (MOCS and MOFPA) |
| $n_{CS}$ | Cuckoo population (MOCS) |
| $n_{FPA}$ | Pollen population (MOFPA) |
| $p$ | Switch probability between local and global pollination (MOFPA) |
| $s$ | Step size of Lévy flight (MOFPA) |
| $t$ | Current iteration of the optimization algorithm (MOCS and MOFPA) |
| $u$ | Lévy distribution approximation, for the MOCS algorithm |
| $w_i$ | Weight of single objective i, in a multi-objective function |
| $x_i$ | Data points to clusterize (K-means) |
| $z_{ik}$ | Cluster membership variable (K-means) |
| $\alpha$ | Step size of Lévy flight (MOCS) |

| Table 1 (continued) | |
|---|---|
| **Symbols** | **Descriptions** |
| $\beta$ | Random step length (MOCS) |
| $\varepsilon$ | Parameter for local pollination strength (MOFPA) |
| $\lambda$ | Parameter for global pollination strength (MOFPA) |

In order to the utilize multi-objective functions, the simplest way is to involve all objectives into a single mathematical sentence, as shown in Eq. (1):

$$f = \sum_{i=1}^{n} w_i f_i, \quad \sum_{i=1}^{n} w_i = 1, \tag{1}$$

in which $w_i$ are the weights given to each objective, and $f_i$ are the single objectives and the sum of all weights must be equal to 1.

## Multi-objective cuckoo search

One of the most utilized bioinspired algorithms, the Cuckoo Search algorithm has been created by *Yang & Deb (2009)*. It has been used to optimize multiple applications ever since, mainly in the areas of mathematics and engineering, according to *Shehab, Khader & Al-Betar (2017)* and *Ferreira et al. (2021)*. The main bioinspired idea behind this method is the computational modeling of the parasitic behavior of cuckoo-type birds, that often lay their eggs inside the nests of other types of birds. This is a natural occurrence in nature, as the host species often does not perceive the cuckoo's "alien" egg inside the nest, or either choose to ignore it completely or abandon the nest altogether, choosing another place to lay its eggs on. However, if the egg is ignored and left to grow inside the host bird's nest, the cuckoo hatchling is born, and reaches maturity much faster than the other eggs, pushing the host bird's eggs outward. Thus, the cuckoo baby bird expels the other eggs from the nest, resulting in a higher food share for it, and becoming well-fed.

Thus, the whole process is based upon three major rules:

1. Each cuckoo lays one egg at a time, and deposits it in a random nest. Each egg is considered a potential solution—metaheuristic

2. The best nests carry the best eggs (solutions), and these will survive the next generations due the parasitic nature of the cuckoo hatchlings—elitism.

3. The number of available nests is constant, and defined by the code developer. The probability of a cuckoo egg being discovered by the host bird is defined as $Pa \in [0, 1)$. After this, the bird may choose to discard this egg or abandon its nest—discarding the worst solutions.

The latter rule can also be described by indicating that a probability fraction Pa from the various n nests of host birds are replaced by new nests, presenting randomized solutions.

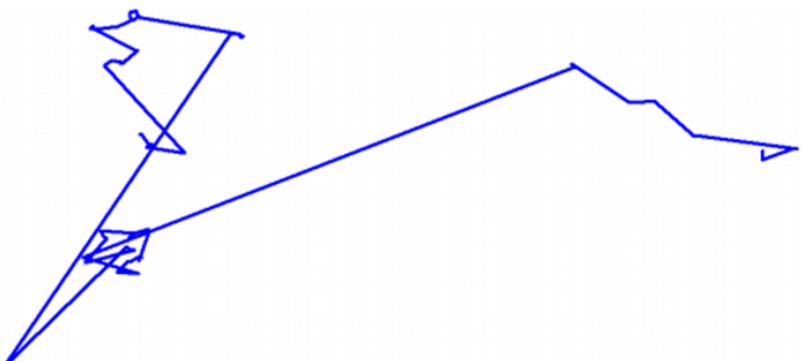

**Figure 2** Representation of a 100-step Lévy flight in its search space.

The movement of cuckoos within the search space is dictated by a stochastic method called Lévy flights, which provides each cuckoo (from a number of i cuckoos) in the code iteration a path to find possible solutions (nests). Equations (2) and (3) represent, mathematically, a Lévy flight and its Lévy distribution, respectively:

$$X_i^{t+1} = X_i^t + \alpha \oplus Levy(\beta) \tag{2}$$

$$Levy(\beta) \cong u = t^{-(\beta+1)}; (1 < \beta \leq 3) \tag{3}$$

where i and t are, respectively, the maximum number of cuckoos and the current code iteration. $\alpha$ is a parametric value denoting the step size for the cuckoo flight, that must generally be greater than zero. In this study the value is set to $\alpha = 1$, as *Yang & Deb (2009)* claim that an unitary alpha is sufficient for most optimization cases.

In Eq. (2), the Lévy distribution is associated to the Lévy flight with the product $\oplus$, which means "entrywise multiplication"—a kind of product between two matrices of the same size. In the PSO algorithm, a similar product can be found, however, for the Lévy flight method, the search space can be much better harnessed.

As for Eq. (3), this is about the Lévy distribution. It possesses infinite variance and average values. The variable $\beta$ is the random step length, needed for providing a variable magnitude to the random walk performed by the Lévy flight.

Given that the Lévy distribution presents infinite variance, the search space is virtually limitless, meaning that the length of the flight taken could be very short or incredibly long. However, generally, the new solutions are generated through the Lévy method around the best obtained solution on a given instant, accelerating the process and concentrating the computational effort in a part of the search space. Oftentimes, however, solutions are generated randomly across the space—this is good to prevent the algorithm from getting stuck in local optima, which are not the globally best possible solution.

An illustration showing a Lévy flight with around a hundred steps can be found in Fig. 2.

A pseudocode of the CS algorithm, as well as its multi-objective counterpart, can be found in the article in which it was proposed by *Yang & Deb (2009)*. Below, in Algorithm 1, a transcription of this code is presented.

---

**Algorithm 1 Multi-objective cuckoo search algorithm.**

Define the objective functions as $f_n(x), x = (x_1, ..., X_d)^t$

Generate the initial population of $n$ host nests $x_i (i = 1, 2, ..., n)$

**while** ($t <$ number of iterations) or (stop criterion) **do**

    Select a cuckoo randomly *via* Lévy flight (Eq. (2))

    Evaluate the cuckoo's fitness (represented by F)

    Draft a random nest (say, j) out of the $n$ available

    **if** $F_i < F_j$ **then**

        Replace randomly drafted $j$ by the new solution $x_i$

    **end if**

    Discard a fraction $P_a$ of worse nests and build new ones

    Keep the best/better quality solutions

    Rank the best nest set and find the current best

**end while**

Post-process and visualize results

---

## Multi-objective flower pollination

A solution based on the behavior of natural flower pollinators has been proposed by *Yang (2012)*, the same author of the cuckoo search, which also utilizes the Lévy flight method of optimal space search. Its efficiency, in many single and multi-objective applications is proven to be greater than particle swarm and genetic optimization algorithms. Its multi-objective counterpart has been proposed in 2013, also by *Yang, Karamanoglu & He (2013)*.

As for the stages of the algorithm, four rules are defined thusly:

1. Biotic and cross-pollination is considered as global pollination process with pollen-carrying pollinators performing Lévy flights.
2. Abiotic and self-pollination are considered as local pollination.
3. Flower constancy can be considered as the reproduction probability and is proportional to the similarity of two flowers involved.
4. Local pollination and global pollination is controlled by a switch probability p ∈ [0, 1]. Due to the physical proximity and other factors such as wind, local pollination can have a significant fraction p in the overall pollination activities.

Two kinds of pollination are considered and simulated: global and local pollination. This assures that the code does not only fall for local solutions, healthily seeking to encounter a global solution to the objectives. For simplicity manners, the algorithm is based on the idea that every plant possesses only one flower and can pollinate also just one

other flower at a time, when in true biological terms they can hold a few flowers and millions of pollinating gametes. This is so that a plant/flower/pollinating gamete are all considered to be part of one solution altogether.

Hence, the first rule (global pollination) and third rule (flower constancy) of FPA are mathematically represented as shown in Eq. (4):

$$X_i^{t+1} = X_i^t + L(X_i^t - g*) \tag{4}$$

where $X_i^t$ is the pollen i or solution vector $X_i$ at iteration t, and $g^*$ is the current best solution among all solutions at the current iteration.

Pollination strength L is dealt *via* Lévy flight, in which it is a measure of each flight's step size, as denoted in Eq. (5). Here, the flights symbolize the path of insects and pollinator animals in a given area—in the algorithm, this area is the optimization's global search space. However, the equation used in this algorithm differs from the one found in cuckoo flights, as it is based on producing Lévy flights *via* the Mantegna algorithm. This is basically a technique to generate pseudo-random step sizes *via* normal distributions in order to provide an optimal performance whilst still maintaining the demands of the Lévy distribution.

$$L \approx \frac{\lambda \Gamma(\lambda) \sin(\pi \lambda / 2)}{\pi s^{1+\lambda}}, \tag{5}$$

in which $\Gamma(\lambda)$ is the standard, classic gamma function found in Lévy flights, and other probabilistic and complex number applications.

The Mantegna step size algorithm can be explained as the Eq. (6).

$$s = \frac{U}{|V|^{\frac{1}{\lambda}}}, \tag{6}$$

with s being the step size, U being drawn from a Gaussian distribution of variance $\sigma^2$ and V also being drawn from a Gaussian distribution but with unitary variance, as can be verified in *Yang, Karamanoglu & He (2014)*. Generally, the lambda is treated as a parametric value and it is safe to assume that it is a constant with a possible value of around $\lambda \in [0.5, 1.5]$. When $\lambda = 1$, the variance also equals 1, and results are in such case easier to predict.

For the purpose of Rule 2 (local pollination), the flower constancy is mimicked for a limited neighborhood near to the reproductive flower's position. It is represented as

$$X_i^{t+1} = X_i^t + \varepsilon(X_j^t - X_k^t), \tag{7}$$

where $X_j^t$ and $X_k^t$ are pollens from different flowers of the same plant species.

The fourth rule is a probabilistic switch between global and local pollination, and the probability p can be parametrically and singularly adjusted to improve optimization performance, depending on the need of the objective function.

All stages of the algorithm are represented in pseudocode form by the recommendations in *Yang (2012)*, which are transcribed in Algorithm 2. Some details previously discussed can be noticed, such as an if/else switch for global and local pollination, which are done by Lévy flights and random selection, respectively.

---

**Algorithm 2** Multi-objective flower pollination algorithm.

Define the objective functions as $f_n(x), x = (x_1, ..., X_d)^t$

Generate an initial population of flowers/pollens as random solutions

Find, within this population, the best solution $g_*$

Define the switch probability $p \in [0, 1]$

**while** ($t <$ number of iterations) or (stop criterion) **do**

    **for** all $n$ flowers in the population **do**

        **if** *rand* $< p$ **then**

            Generate a step vector $L$ which obeys a Lévy distribution

            Execute global pollination as in Eq. (4)

        **else**

            Pick a uniformly distributed number from $\varepsilon = [0, 1]$

            Randomly choose individuals $j$ and $k$ from all solutions

            Do local pollination according to Eq. (7)

        **end if**

        Evaluate new solutions

        Update solutions that are better into the population

    **end for**

    Find the best current solution, represented by $g_*$

**end while**

Post-process and visualize results

---

## The K-means algorithm

An unsupervised learning method of computational intelligence, the K-means clustering algorithm has the goal of grouping similar data points. Each of those $k$ amount of clusters, then, possesses a centroid, which is the mean value M of all positions of the data points. Due to being fast and easy to reproduce, it is one of the most popular clustering algorithms. *Sinaga & Yang (2020)* defines the objective function of K-means according to the following equation (Eq. (8)):

$$J(A, Z) = \sum_{i=1}^{n} \sum_{k=1}^{c} z_{ik} ||x_i - a_k||^2 \tag{8}$$

in which $x_i$ is a data point i belonging to a dataset $X = \{x_1, x_2, ..., x_n\}$ spread over an Euclidean space $\mathbb{R}^d$ of d-dimensions, $a_k \in A = \{a_1, a_2, ..., a_n\}$ is the centroid of the k-th

cluster and $z_{ik}$ is a binary variable (either 0 or 1) that signals if the user i belongs to cluster k. Given that the objective function needs to be minimized, the algorithm then proceeds to alter the centroids by also minimizing the Euclidean distance between them and the data points that belong to them, according to Eq. (9):

$$a_k = \frac{\sum_{i=1}^{n} z_{ik} x_i}{\sum_{i=1}^{n} z_{ik}}, z_{ik} = \begin{cases} 1, & \text{if } ||x_i - a_k||^2 = \min_{(1 \leq k \leq c)} ||x_i - a_k||^2. \\ 0, & \text{otherwise.} \end{cases} \tag{9}$$

where $||x_i - a_k||^2$ is the Euclidean distance.

## The ECC-33 propagation model

A channel path loss model is required to evaluate which users are connected to the network. This propagation model should both belong to a frequency range where the 5G-NR channels are going to be propagated, as well as adapt to the environment in which it shall be applied. Some examples in the literature by *Samad, Choi & Choi (2022)* and *Oladimeji, Kumar & Elmezughi (2022)* address models such as the Close-In (CI), Floating Intercept (FI) and Alpha-Beta-Gamma (ABG), along with measurement campaigns to verify their usage in 5G channel modelling characteristics.

However, in this study, an ECC-33 model was chosen, which is adapted to the urban but forested environments of the city of Belém, as proposed by *de Carvalho et al. (2021)*.

ECC-33 is derived from the famous Okumura-Hata model, but taking into consideration the behavior of higher frequencies, thus extending the frequency range. In *Mollel & Michael (2014)*, experimental results have been drawn by comparing different path loss models conducted in Dar es Salaam, Tanzania. The COST-231 and ECC-33 models had the lesser Root Mean Square Error (RMSE), meaning that—for urban, suburban and rural settings—they had the most approximate results to measured signal data.

Therefore, the model used in this work is defined thusly:

$$PL_{ECC} = A_{fs} + A_{bm} - GF_t - GF_r \tag{10}$$

in which $A_{fs}$ is the free-space attenuation, $A_{bm}$ is the base median path loss and $GF_t$ and $GF_r$ are the transmitting antenna gain factor and received antenna gain factor, respectively. The values used for the model correspond to the large city scenario, as all acquired data correspond to an urban region of medium-to-high density. The following Eqs. (11)–(14) further detail the variables of the model:

$$A_{fs} = 92.4 + 20log(d) + 20log(f_c) \tag{11}$$

$$A_{bm} = 20.41 + 9.83log(d) + 7.89log(f_c) + 9.56(log(f_c))^2 \tag{12}$$

$$GF_t = log\left(\frac{h_{BS}}{200}\right)(13.958 + 5.8log(d))^2 \tag{13}$$

$$GF_r = 0.759(h_{UE}) - 1.862 \tag{14}$$

in which $f_c$, $h_{BS}$ and $h_{UE}$ are the central frequency, the base-station antenna height and the user device's antenna height, respectively. While $d$ is the distance between a user and a

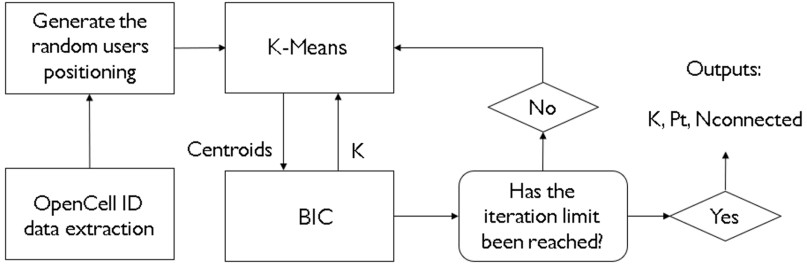

**Figure 3** **Flowchart of the processes involved in this study.**

base-station, that in this study is calculated by the haversine distance formula, as it appears on *Gade (2010)* and *Chopde & Nichat (2013)*:

$$d = 2R \arcsin \sqrt{sin^2\left(\frac{\mu_2 - \mu_1}{2}\right) + cos(\mu_1)cos(\mu_2)sin^2\left(\frac{\lambda_2 - \lambda_1}{2}\right)} \tag{15}$$

where $\lambda$ are the latitudinal positions, $\mu$ are the longitudinal positions, and $R$ is the approximate radius of the Earth ($6.371.10^6$ m). Furthermore, the received power for an user is represented in Eq. (16):

$$P_r = P_t - PL_{ECC} + G_t + G_r, \tag{16}$$

in which $G_t, G_r$ are the transmitting (Tx) antenna gain and receiving (Rx) antenna gain.

## The hybridization process

In real-life scenarios, not often is the number of clusters to achieve optimal results is known beforehand. For that, there are some heuristic methods that can be utilized to get approximations of how many clusters one might need to satisfy an application. For instance, *Khan et al. (2022)* have chosen the Elbow heuristic. However, given that one might need a more complex and sophisticated way of deciding how to cluster, and how much to cluster, like in a scenario of network dimensioning and coverage area optimization, these heuristics may not provide optimal solutions. This is why a metaheuristic method, like a bioinspired algorithm, is more suited for this task. The hybridization of a clustering algorithm like K-means and a metaheurisitc BIC means that both of those methods benefit from each other. The K-means provides the base-station coordinates on a given area, and the BIC provides the values of the ideal number of clusters, taking into consideration multiple inputs that will decide how much clustering there must be. Figure 3 demonstrates the whole process of the study. Please notice how the K-means processes a geographical position for cluster centers and the BIC returns it a number of clusters to attempt and see if an optimal result is reached. This process has the goal, in our study, to provide an optimal value of k clusters, as well as a satisfactory transmitted power value.

A pseudocode displaying the structure of the hybridization can be found in Fig. 3. For each iteration of the BIC, the K-means will run alongside it, providing cluster center (or centroid) values that the BIC shall take and calculate the received power for each and every

---

**Algorithm 3  The hybrid K-means + BIC algorithm.**

Define the constant values of $h_{BS}, h_{UE}, f_c$ and antenna gains

Initialize one of the bioinspired optimization techniques (MOCS or MOFPA)

Generate an initial population with random solutions for $k$ clusters and $P_{BSt}$ transmitted power

**while** (Iterations ≤ Max Iterations) **do**

  Initialize the K-means for each $k$ given by the BIC

  **for** (all the clusters $k$ in the K-means) **do**

    **for** (all the users associated to cluster $k$) **do**

      Calculate the distance between the user and its clusters' centroid *via* (Eq. (15))

      Calculate the ECC path loss, present in Eq. (10)

      Calculate the received power $P_r$ as in Eq. (16), with $P_t$ given by the BIC

      **if** ($P_r \geq -90$ dBm) **then**

        Count the user as within the coverage area

      **end if**

    **end for**

  **end for**

  Calculate the percentage of connected users in relation to the total number of users

  Evaluate, with the BIC, the outputs in a single objective function

  Update the best solutions into the population

**end while**

Post-process and visualize results

---

user, one cluster at a time. If the user has a received power of −90 dBm or greater, it will be considered as connected to the network and, thus, covered.

# RESULTS

A total of 1,900 users have been randomly generated within the simulated area where the small cells shall actuate. This is to ensure that there are users in every corner of the area, and to assure that the clusters can manage to provide coverage for them.

It is worth to denote that the received power threshold to consider a user as connected to its respective cluster is $P_r \geq -90$ dBm. This value has been chosen as a reasonable target for 5G mobile, heterogeneous networks to main good signal strength and capacity.

**Table 2 Values of interest for the simulations.**

| Parameters | Values |
| --- | --- |
| Tx antenna height ($h_{BS}$) | 40 m |
| Rx antenna height ($h_{UE}$) | 1.6 m |
| Tx antenna gain | 3 dBi |
| Rx antenna gain | 0 dBi |
| Central frequencies ($fc$) | [700 MHz, 2.3 GHz, 3.5 GHz] |
| Number of clusters (k) bounds | 1 to 100 clusters |
| Transmitted power ($P_t$) bounds | 30 to 40 dBm |
| Discard probability, for MOCS ($P_a$) | 0.25 |
| Switch probability, for MOFPA ($p$) | 0.5 |

Furthermore, as per stated in *Ayad et al. (2022)*, values of between $-100 \leq P_r \leq -80$ dBm are acceptable for this kind of application.

The objective function for the simulations can be defined as:

$$f = 0.7 N_{outage} + 0.3 P_{diff}, N_{outage} = \frac{N_{users} - N_{connected}}{N_{users}}, P_{diff} = P_t - 30, \qquad (17)$$

in which $P_{diff}$ is the difference of transmitted power to the ideal minimum value of 30 dBm, $N_{users}$ is the total number of users to be attended by the network, $N_{connected}$ are the number of users connected to the small cells and $N_{outage}$ is the percentage of users which are experiencing outage (*i.e.*, disconnected from any cell). Therefore, the main objectives are to minimize the number of users that are excluded from the network whist maintaining the lowest transmitted power possible, that is, $N_{outage} = 0$ and $P_{diff} = 0$.

The search space bounds on the number of clusters and transmitting power are: $L_b = (1 \text{ cluster}, 30 \text{ dBm})$ and $U_b = (100 \text{ clusters}, 40 \text{ dBm})$. These power values for $P_t$ are the limits generally found in small cells, as stated by *Khan, García-Armada & Escudero-Garzás (2020)*. So, the objective for $P_t$ is to approach its minimum value, that is, $P_t = 30$ dBm. It also can be noticed that, since the main goal of the study is to achieve better coverage for users in 5G networks, there is a 0.7–0.3 weight towards coverage over transmitted power optimization in the objective function (Eq. (17)).

Table 2 refers to the constant and variable values that have been coded into the algorithms and the propagation model:

The simulation and its algorithms have been coded in MATLAB, and are run on a personal computer with these characteristics: AMD Ryzen 5 CPU with 4.2 GHz clock and 16 GB DDR4 RAM.

## Results for 700 MHz

Firstly, Figs. 4 and 5 display the results achieved for the 700 MHz frequency band for, respectively, the MOCS-KM and MOFPA-KM. This is a band that possesses a fairly low

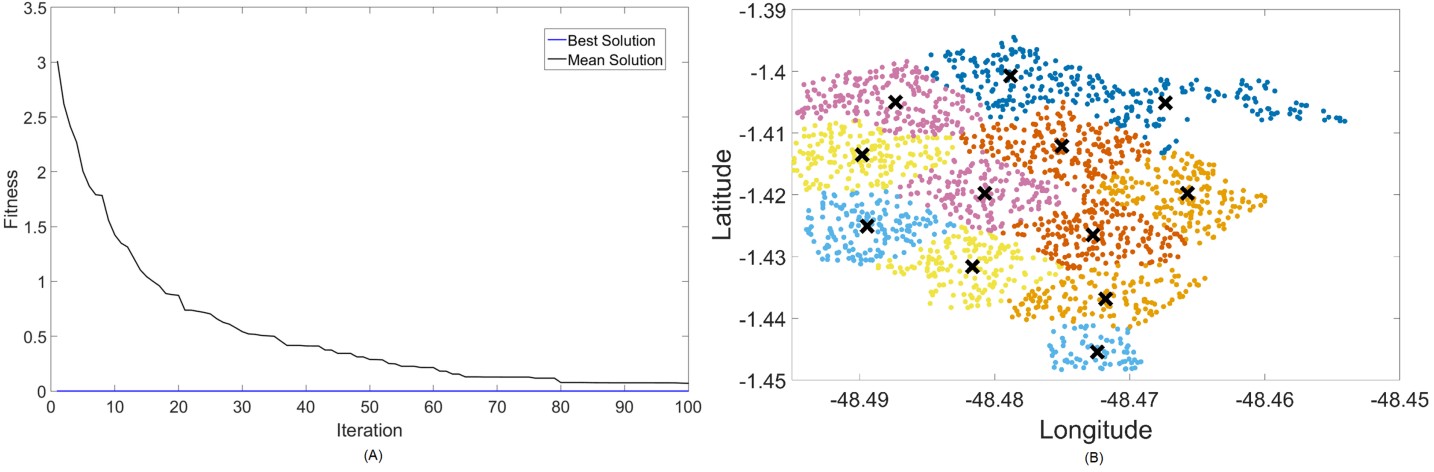

**Figure 4** Results for 700 MHz: (A) fitness plot of MOCS-KM and (B) cluster formation given by the technique.

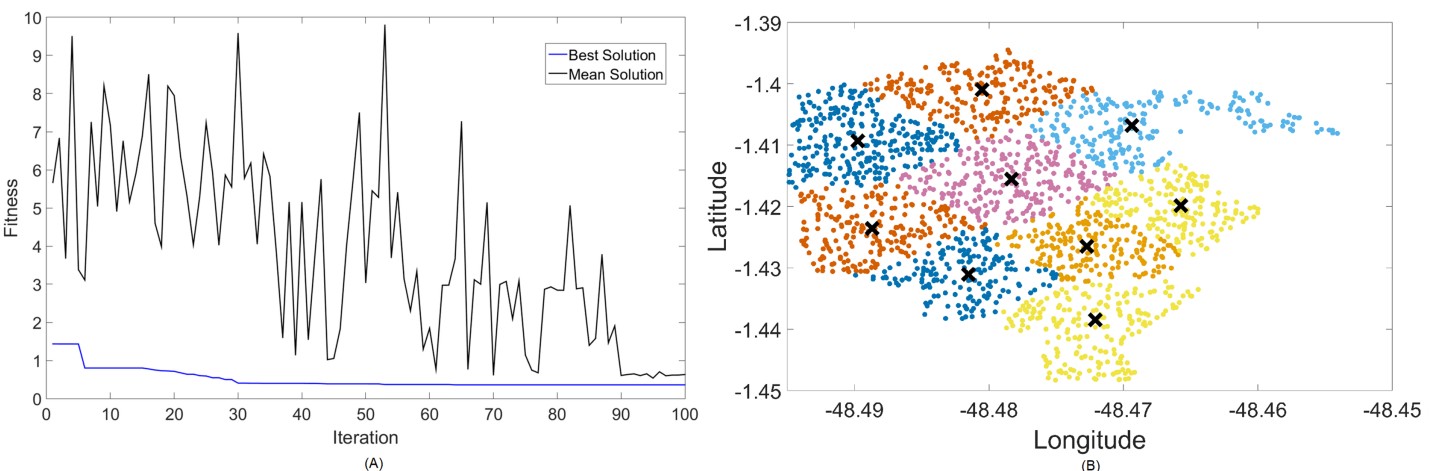

**Figure 5** Results for 700 MHz: (A) fitness plot of MOFPA-KM and (B) cluster formation given by the technique.

data rate and bandwidth, but since it is one of the auctioned bands for 5G operation in Brazil, it is included here as means of comparison and to test the BIC + K-means hybrids.

The X symbols in the figures displaying the clusters are referent to the position of centroids, and the colored dots correspond to the users.

Here, the algorithm had no problems whatsoever in optimizing for zero fitness, needing only a few iterations to converge. This is because path loss is less intense for lower frequencies, and so a small group of small cells can already provide perfect coverage.

The best solver in this case is MOCS-KM, with a solution of 12 clusters and power of 30 dBm. However, MOFPA-KM comes very close to a zero fitness value as well, with an inferior number of clusters of only 9 and 30 dBm power.

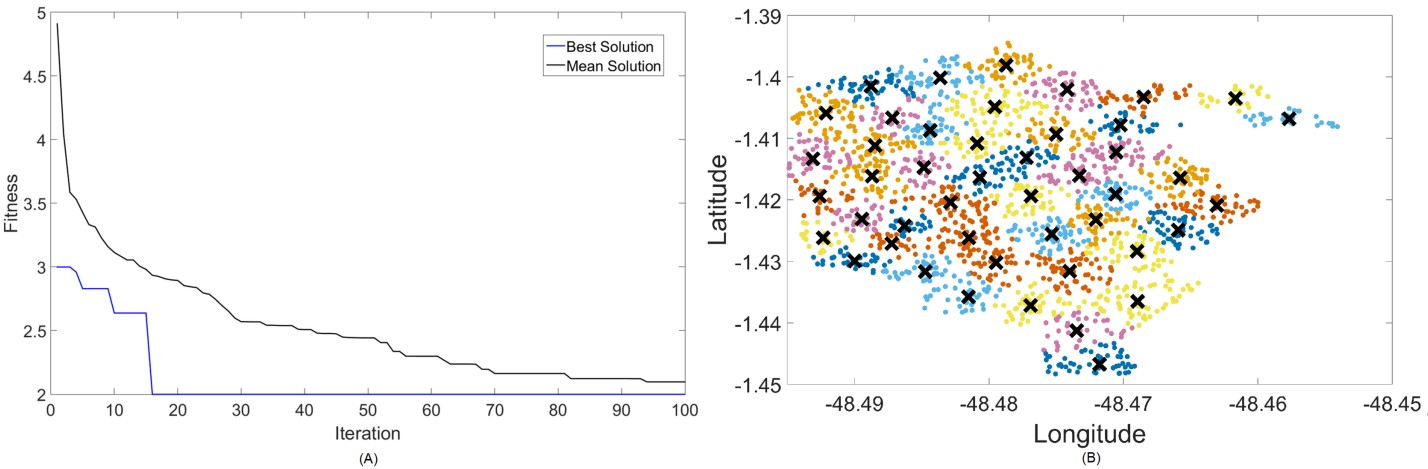

**Figure 6** Results for 2.3 GHz: (A) fitness plot of MOCS-KM and (B) cluster formation given by the technique.

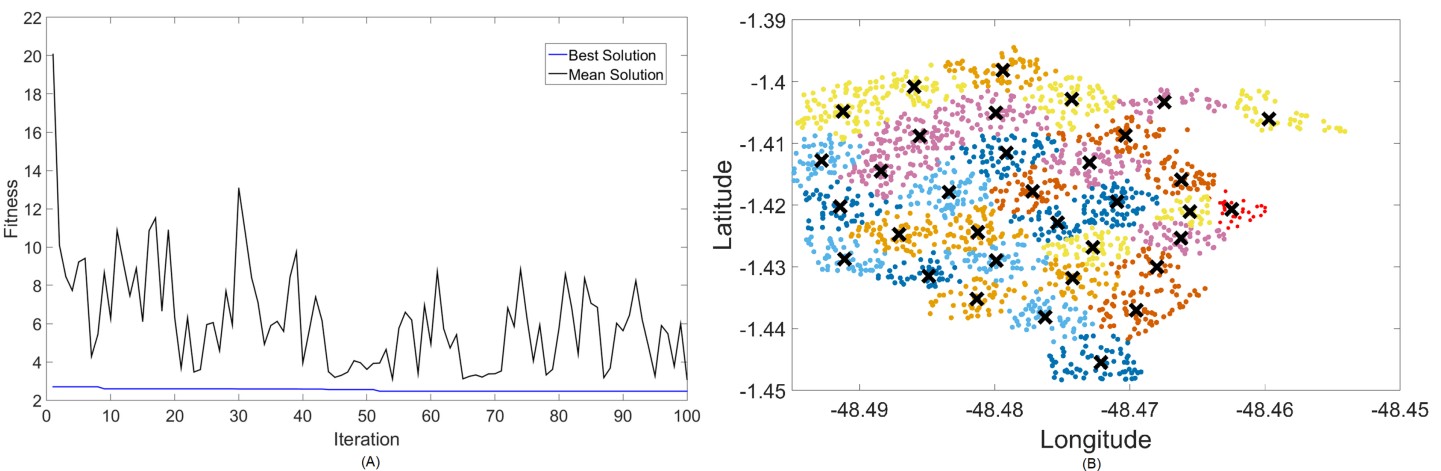

**Figure 7** Results for 2.3 GHz: (A) fitness plot of MOFPA-KM and (B) cluster formation given by the technique.

### Results for 2.3 GHz

The band that promises to function both in LTE and in 5G is shown next. Figures 6 and 7 are the results for the 2.3 GHz frequency band. It is expected to operate in smaller cities as a means of digital inclusion, providing more bandwidth to 4G-LTE demands, or in urban centers as a complementary traffic alternative to the main band of 5G, which is 3.5 GHz.

Results for this band are already harder to solve than 700 MHz, with the best fitness results coming from MOCS-KM once again. It has achieved $f_{best} = 2$, with 46 clusters and a satisfactory transmitted power of 31 dBm. MOFPA-KM has proposed a higher $P_t$ value with less clusters, resulting in $f_{best} = 2.4267$, $k = 34$, and $P_t = 33.3352$ dBm.

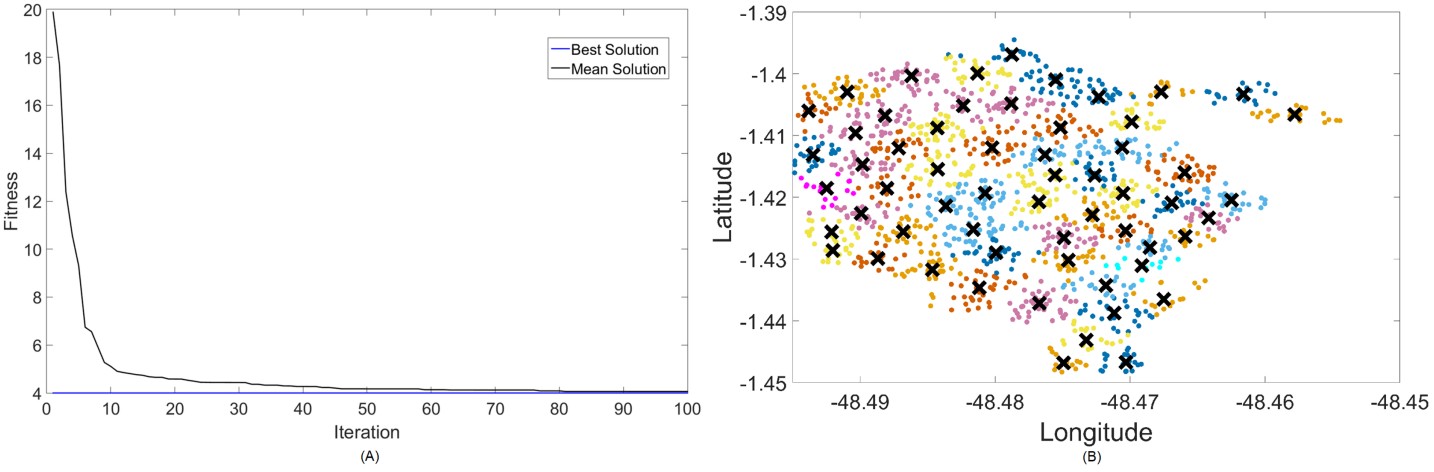

**Figure 8** Results for 3.5 GHz: (A) fitness plot of MOCS-KM and (B) cluster formation given by the technique.

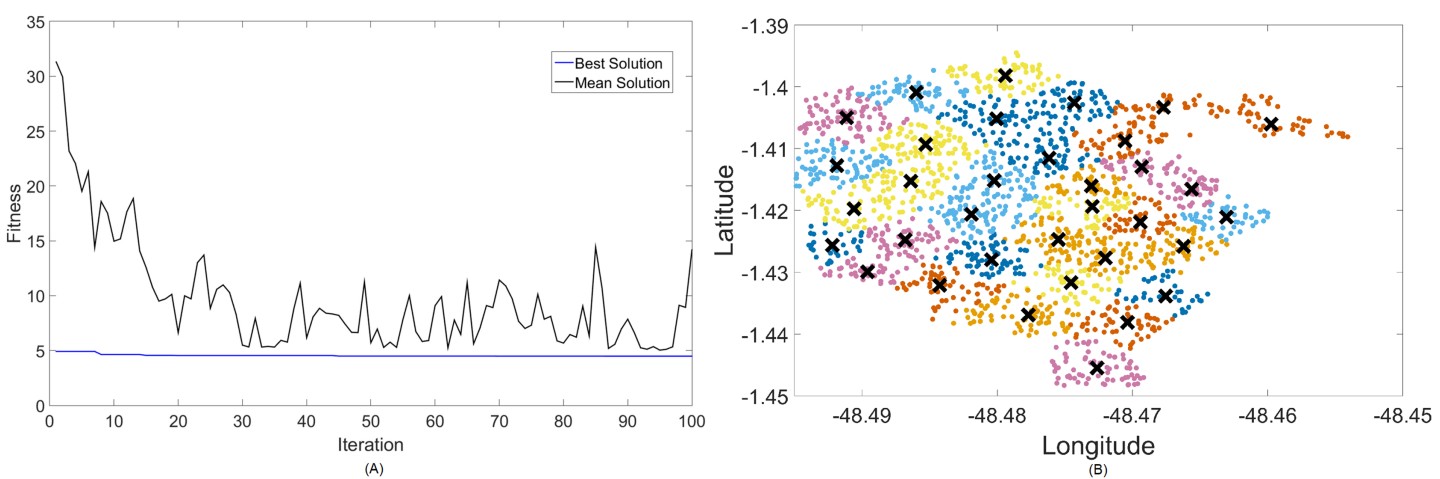

**Figure 9** Results for 3.5 GHz: (A) fitness plot of MOFPA-KM and (B) cluster formation given by the technique.

## Results for 3.5 GHz

Finally, the main and most used frequency band of 5G. The 3.5 GHz spectrum possesses up to 80 MHz of band per operator in Brazil, and is set to expand in highly-urbanized areas, as it can handle greater traffic and provide faster data rates than the other two. Figures 8 and 9 demonstrate the results of the simulation for this frequency.

For this case, the results are basically the same in terms of coverage, but fitness values turn greater because of the choice in both algorithms to utilize more transmitted power instead of increasing the number of clusters too much. In terms of fitness, MOCS-KM marginally wins again, with $f_{best} = 4.006$. That being said, MOFPA-KM has converged to a solution with 34 clusters again—the same number from the 2.3 GHz simulation. It has,

**Table 3 Results for each best iteration of the hybrid techniques.**

| Simulation | $k$ | $P_t(dBm)$ | $N_{outage}(\%)$ | $P_{diff}(dBm)$ | $f_{max}$ | $f_{avg}$ | $f_{best}$ | Time (s) |
|---|---|---|---|---|---|---|---|---|
| 700 MHz (MOCS-KM) | 12 | 30 | 0 | 0 | 0.7892 | 0.1327 | 0 | 552 |
| 700 MHz (MOFPA-KM) | 9 | 30 | 0.5133 | 0 | 1.9397 | 0.6075 | 0.3593 | 594 |
| 2.3 GHz (MOCS-KM) | 46 | 31 | 2.4285 | 1 | 10.9882 | 2.1649 | 2 | 848 |
| 2.3 GHz (MOFPA-KM) | 34 | 33.3352 | 2.0373 | 3.3352 | 4.7515 | 2.7564 | 2.4267 | 843 |
| 3.5 GHz (MOCS-KM) | 59 | 38 | 2.2857 | 8 | 5.5686 | 4.1163 | 4.006 | 993 |
| 3.5 GHz (MOFPA-KM) | 34 | 40 | 2.0421 | 10 | 6.3954 | 4.7312 | 4.4295 | 905 |

however, increased the transmitted power to its maximum value of 40 dBm in order to achieve that.

## Table of results

In Table 3, results for the optimization processes of MOCS-KM and MOFPA-KM hybrids are shown. For the best generated population (the one which produced $f_{best}$), it records the maximum, medium and best fitness values, as well as the outputs and the running time of the respective simulations.

The parameters and variables displayed in Table 3 are: the optimal number of clusters (k); optimal transmitted power per cluster ($P_t$, in dBm); the percentage of users suffering outage ($N_{outage}$, in %); the differential between optimal $P_t$ and the lower bound value of $P_t = 30$ dBm ($P_{diff}$, in dBm); maximum objective function fitness value found within the best generation ($f_{max}$); average fitness value of all individuals in the best generation ($f_{avg}$); the fitness value of the best individual, in the best generation ($f_{best}$); and the total running time of the optimizations (Time, in seconds).

## Computational analysis

In order to elucidate the performance of both algorithms, a numerical analysis of their computational complexity has been conducted.

Theoretically, the study of algorithmic complexity of bioinspired computing and clustering hybrids has already been conducted. *Kaur, Pal & Singh (2020)* denotes that the complexity of both CS-KM and FPA-KM hybrids are set to linear variables in the Big-O formula—see *Bae & Bae (2019)*. These are still valid for multi-objective counterparts of the same techniques. Thus, the equations for algorithmic time complexity, adapted from *Kaur, Pal & Singh (2020)*, are as follows:

$$O_{MOCS-KM} = O(N_i * k * n_{CS} * N_{users}) \tag{18}$$
$$O_{MOFPA-KM} = O(N_i * k * n_{FPA} * N_{users}) \tag{19}$$

in which $N_i$, $k$, $n_{CS}$, $n_{FPA}$ and $N_{users}$ are the number of iterations in the code, the number of clusters, the cuckoo population (for MOCS), the pollen population (for MOFPA), and the number of users to be covered by the network, respectively. The area of coverage, as considered in the original formula, can be replaced into the number of users to be covered

**Peer**J Computer Science

**Table 4 Parameters and values used for the numerical analysis.** In italic: values kept constant between simulations.

| Parameters | Values |
| --- | --- |
| Iterations ($N_i$) | [50, *100*, 150, 200] |
| Clusters (k) | [1 to 50, *1 to 100*, 1 to 150, 1 to 200] |
| Cuckoo population ($n_{CS}$) | [15, 20, *25*, 30] |
| Pollen population ($n_{FPA}$) | [15, 20, *25*, 30] |
| Users ($N_{users}$) | [400, 900, 1,400, *1,900*] |

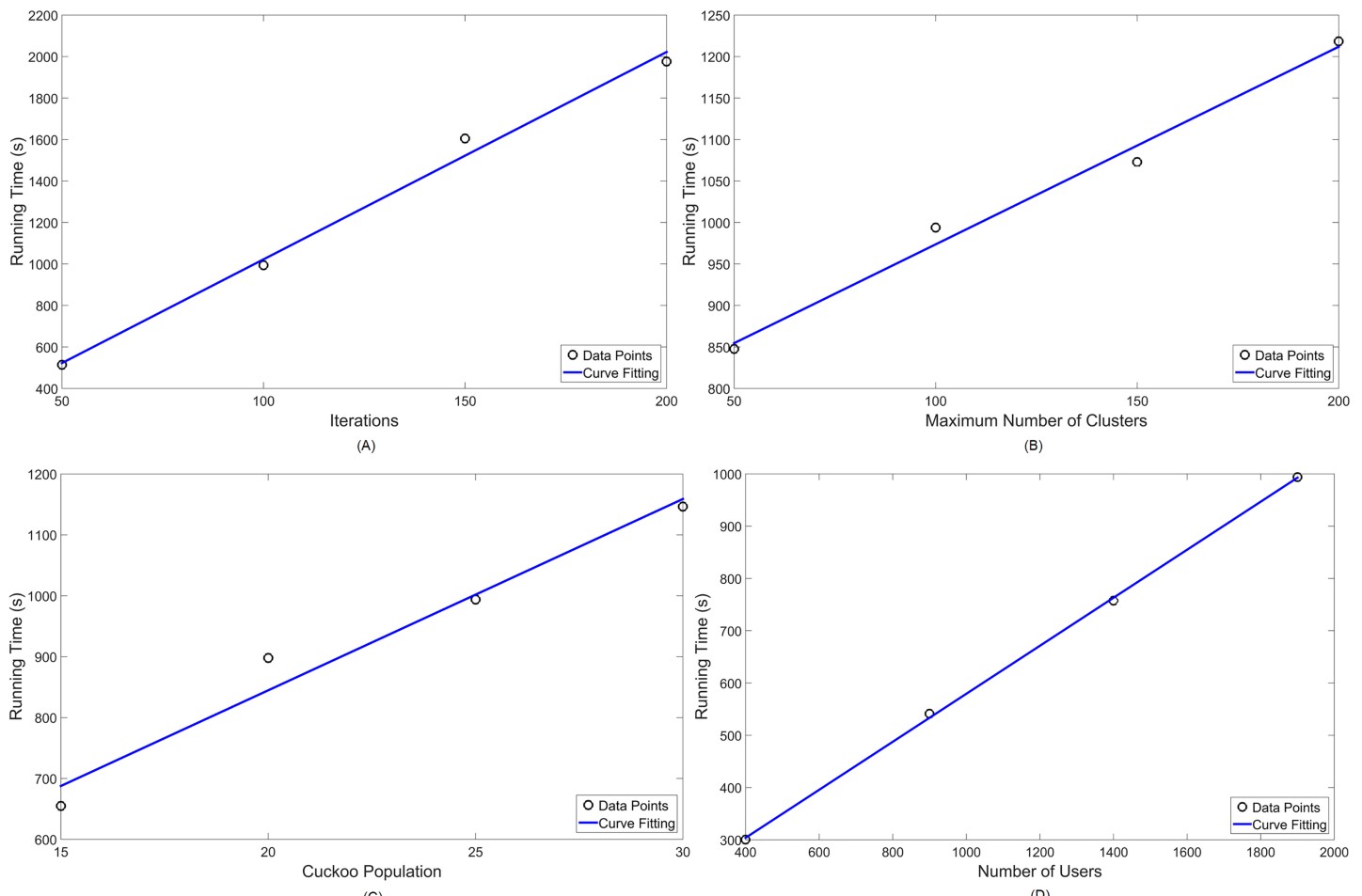

**Figure 10 Numerical analysis over running time for MOCS-KM: (A) number of iterations ($N_i$); (B) number of clusters (k); (C) Cuckoo population ($n_{CS}$) and (D) number of users ($N_{users}$) in the simulation.**

by the network. In general terms, the more users a network needs to cover, the greater the area of coverage should be.

Numerical simulations have been performed taking into account the running time of the algorithms. Therefore, a set of 32 data points, 16 per technique, were obtained *via* simulations in order to measure the linearity of computational complexity. That is because,

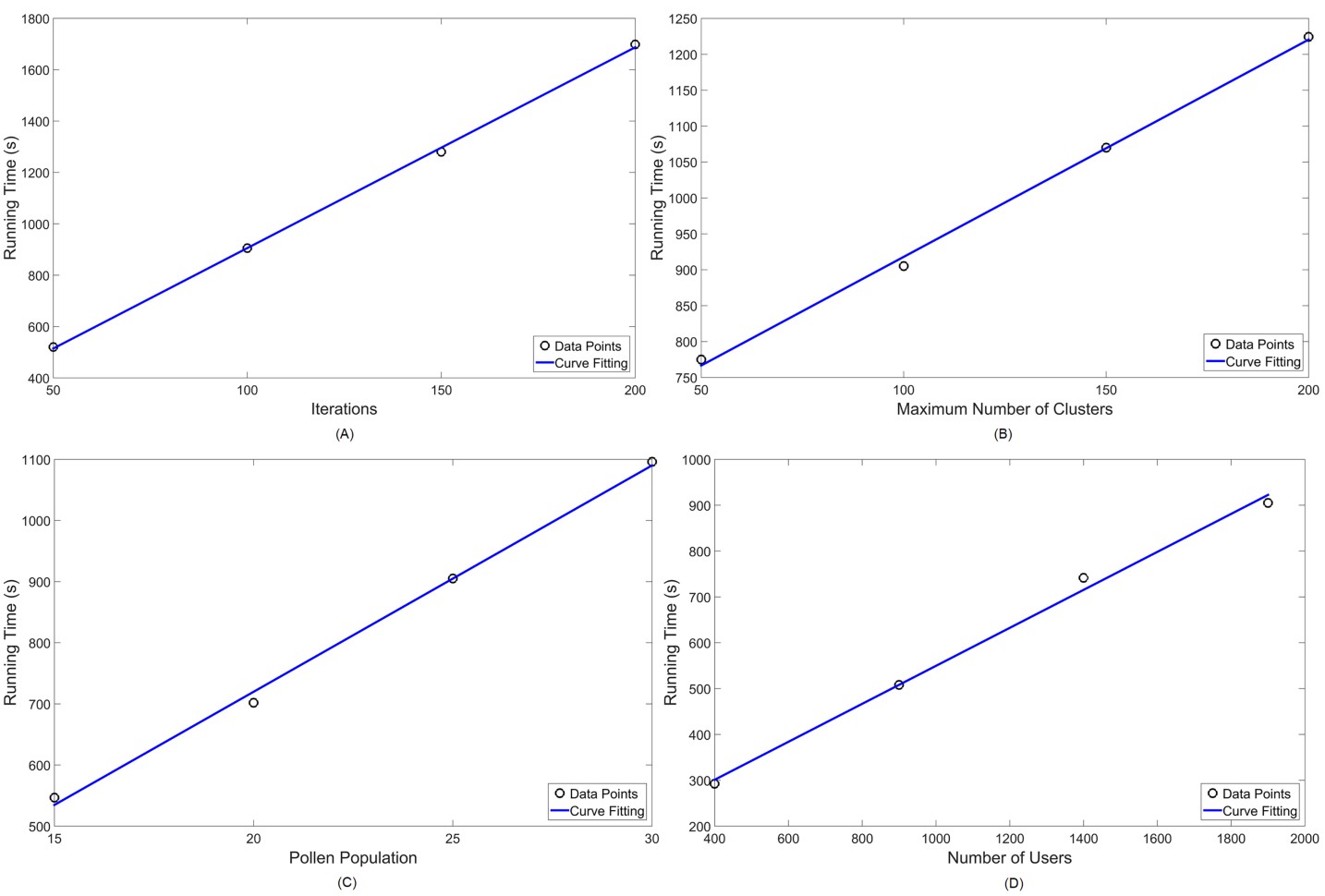

**Figure 11 Numerical analysis over running time for MOFPA-KM: (A) number of iterations ($N_i$); (B) number of clusters (k); (C) pollen population ($n_{FPA}$) and (D) number of users ($N_{users}$) in the simulation.**

for each simulation, one of four parameters (iterations, clusters, population and users) is chosen to vary between four values, while all others are kept to fixed values.

*Cardoso et al. (2020)* is an example of an article that provides a numerical analysis of results. Although it utilized another network planning method (cellular automata, or CA), its methodology, along with *Kaur, Pal & Singh (2020)*, have served as basis for the computational analysis made in this work.

Table 4 shows the variations chosen for each parameter, denoting in italic letters which ones are kept constant—for both MOCS-KM and MOFPA-KM. Lastly, Figures 10 and 11 display the numerical analysis results for MOCS-KM and MOFPA-KM parameters, respectively. These are curve fittings (blue line) taking into account the simulated data points (circles) in MATLAB.

Given that simulational results have shown in Table 3 to operate, in standard parameter values, in around 10 to 15 min, these are satisfactory approximations for the NP-hard

**Table 5 Running times and coverage ratio of MOCS-KM for different data points.**

| $N_i$ | $k$ | $n_{CS}$ | $N_{users}$ | Running time (s) | Coverage ratio (%) |
|---|---|---|---|---|---|
| 50 | [1..100] | 25 | 1,900 | 513.517 | 96.1 |
| 100 | [1..100] | 25 | 1,900 | 993.888 | 97.4 |
| 150 | [1..100] | 25 | 1,900 | 1,604.884 | 97.4 |
| 200 | [1..100] | 25 | 1,900 | 1,976.066 | 97.4 |
| 100 | [1..50] | 25 | 1,900 | 847.608 | 97.4 |
| 100 | [1..100] | 25 | 1,900 | 990.723 | 97.4 |
| 100 | [1..150] | 25 | 1,900 | 1,072.863 | 97.4 |
| 100 | [1..200] | 25 | 1,900 | 1,218.268 | 97.4 |
| 100 | [1..100] | 15 | 1,900 | 654.813 | 97.4 |
| 100 | [1..100] | 20 | 1,900 | 897.975 | 97.4 |
| 100 | [1..100] | 25 | 1,900 | 991.452 | 97.4 |
| 100 | [1..100] | 30 | 1,900 | 1,146.521 | 97.4 |
| 100 | [1..100] | 25 | 400 | 654.813 | 95 |
| 100 | [1..100] | 25 | 900 | 897.975 | 96 |
| 100 | [1..100] | 25 | 1,400 | 991.452 | 96.6 |
| 100 | [1..100] | 25 | 1,900 | 1,146.521 | 97.4 |

**Table 6 Running times and coverage ratio of MOFPA-KM for different data points.**

| $N_i$ | $k$ | $n_{FPA}$ | $N_{users}$ | Running time (s) | Coverage ratio (%) |
|---|---|---|---|---|---|
| 50 | [1..100] | 25 | 1,900 | 520.203 | 97.2 |
| 100 | [1..100] | 25 | 1,900 | 905.090 | 98 |
| 150 | [1..100] | 25 | 1,900 | 1,279.922 | 98 |
| 200 | [1..100] | 25 | 1,900 | 1,976.066 | 98 |
| 100 | [1..50] | 25 | 1,900 | 774.981 | 95.4 |
| 100 | [1..100] | 25 | 1,900 | 903.691 | 98 |
| 100 | [1..150] | 25 | 1,900 | 1,070.089 | 98 |
| 100 | [1..200] | 25 | 1,900 | 1,224.242 | 98 |
| 100 | [1..100] | 15 | 1,900 | 546.529 | 98 |
| 100 | [1..100] | 20 | 1,900 | 701.901 | 98 |
| 100 | [1..100] | 25 | 1,900 | 908.275 | 98 |
| 100 | [1..100] | 30 | 1,900 | 1,095.929 | 98 |
| 100 | [1..100] | 25 | 400 | 292.472 | 93.8 |
| 100 | [1..100] | 25 | 900 | 508.127 | 96.8 |
| 100 | [1..100] | 25 | 1,400 | 741.414 | 97.5 |
| 100 | [1..100] | 25 | 1,900 | 902.978 | 98 |

problem of cell planning. Furthermore, their performance are linearly scalable, thus easy to predict for a set of parameter values, as seen in Figs. 10 and 11.

Table 5 denotes the running times and coverage capacity (by number of users covered) of data points for MOCS-KM, whilst Table 6 does so for MOFPA-KM, both for the

frequency of 3.5 GHz. These two variables are measured in the analysis of *Cardoso et al. (2020)*, thus it is useful to detail them further here as well. The coverage ratio, which is the amount of users connected to the clusters, can be calculated as $N_{connected}$ taken from objective function (Eq. (17)).

In Table 5, it is noticeable that the maximum coverage achieved by MOCS-KM, considering all users at 3.5 GHz, is capped at 97,4%. A likewise behavior is seen for MOFPA-KM, in Table 6, where maximum capacity is capped at 98%. When reducing the number of users, they may become more scattered in the search space, thus resulting in slightly lower coverage in some cases.

## CONCLUSIONS

Smart cities will bring forth many future challenges, and this work proposed a novel small cell positioning system using a hybrid approach to provide better user coverage and to save more energy. The presented scheme deals with a method of clustering and optimizing for the implementation of 5G small cells according to user traffic, using the city of Belém, Brazil as a simulational example. Optimization was provided by the two hybrid methods, MOCS-KM and MOFPA-KM.

In general the simulations were satisfactory, as user outage is never greater than 2.5% for all cases, making the amount of connected users over 97%. Transmitted power levels have been kept within small cell ranges, providing good implementation opportunities. That being said, the techniques have a sort of preference that differentiate themselves. Also, it can be denoted that MOCS-KM prefers to increase the amount of clusters to reduce the transmitted power, whilst MOFPA-KM is the opposite.

Therefore, the usage of said hybridizations should be a matter of preference for the network planner professional that intends to use them. For coverage optimization both present similar and adequate results, but MOFPA-KM keeps cluster numbers to a bare minimum and optimizes for greater power usage; and MOCS-KM is a bit more precise and uses less power per cluster but produces a higher amount of clustering. So, these variables are bound to limitations and cost of implementation issues that are up to which of those are less costly to produce.

Additionally, an analysis of the performance of both hybrid techniques has been tested in the results. Even though these are complex problems that cannot be solved in real-time, there is a predictable and linear behavior of running time and fitness of results for one-variable variations. It is also attested that the number of iterations in the code has the greatest influence in computational cost. Oftentimes, processing times are lower for MOFPA-KM and so is the coverage ratio.

Further challenges to enrich the study would be to implement more 5G capacity variables into the simulation. The focus of this article has been mostly on coverage problems, but there is much to add in concern to capacity dimensioning. Further opportunities for research involve minimization and measurement of network cost and maximization of throughput of data per area.

### Funding

This work was supported by the Coordination for the Improvement of Higher Education Personnel—CAPES, the National Council for Scientific and Technological Development—CNPq, and the Support Program for Qualified Production—PROPESP/UFPA (PAPQ). The funders had no role in study design, data collection and analysis, decision to publish, or preparation of the manuscript.

### Grant Disclosures

The following grant information was disclosed by the authors:
Coordination for the Improvement of Higher Education Personnel—CAPES.
National Council for Scientific and Technological Development—CNPq.
Qualified Production—PROPESP/UFPA (PAPQ).

### Competing Interests

The authors declare that they have no competing interests.

### Author Contributions

- Flávio Henry Ferreira conceived and designed the experiments, performed the experiments, analyzed the data, performed the computation work, prepared figures and/or tables, authored or reviewed drafts of the article, and approved the final draft.
- Fabrício José Brito Barros conceived and designed the experiments, performed the experiments, analyzed the data, authored or reviewed drafts of the article, and approved the final draft.
- Miércio Cardoso de Alcântara Neto conceived and designed the experiments, performed the experiments, analyzed the data, authored or reviewed drafts of the article, and approved the final draft.
- Evelin Cardoso conceived and designed the experiments, performed the experiments, analyzed the data, authored or reviewed drafts of the article, and approved the final draft.
- Carlos Renato Lisboa  Francês conceived and designed the experiments, performed the experiments, analyzed the data, authored or reviewed drafts of the article, and approved the final draft.
- Jasmine Araújo conceived and designed the experiments, performed the experiments, analyzed the data, performed the computation work, authored or reviewed drafts of the article, and approved the final draft.

### Data Availability

The raw data and code are available in the Supplemental Files.

### Supplemental Information

Supplemental information for this article can be found online at http://dx.doi.org/10.7717/peerj-cs.1412#supplemental-information.

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
