# Peer review of "Hybrid computational and real data-based positioning of small cells in 5G networks"

_PeerJ Computer Science, doi:10.7717/peerj-cs.1412_

## Round 0.1 · original submission · Major Revisions

· Academic Editor

Major Revisions

Both the reviewers have raised a number of concerns which should be addressed.

Reviewer 1 ·

Basic reporting

This paper is about the hybrid computational method based positioning of small cells in 5G networks. The work is interesting and useful for 5G networks. Following are the suggestions to the authors:
1. The title of the paper is long and is in in the form of a sentence. The title should be revised. Some suggestions for the revised titles are ‘Hybrid computational approach-based positioning of small cells in 5G networks’ or ‘Hybrid computational and real data-based positioning of small cells in 5G networks.’
2. In the abstract, in first line, there is a space between last word and full stop. It should be corrected. Similarly, ‘high demand in smart cities. . In this paper’ should be corrected as ‘high demand in smart cities. In this paper’.
3. Authors focus on the sub-6 GHz frequency range of 5G spectrum. Please include details of frequency bands for sub-6GHz frequency bands in the introduction section. The references ‘A Wideband Dual-Polarized Filtering Antenna for 5G Sub-6 GHz Base Station Applications’ and ‘Wideband miniaturized patch radiator for Sub-6 GHz 5G devices’ should be used for sub-6 GHz 5G bands.
4. Please move the figure numbers and figure titles below the figures. Also, please move the table numbers and table titles before the tables.
5. The value of the constant alpha is take as unity. Please justify the selection of the value as unity.
6. Page 6 (line 243): Please correct ‘can be found in 2.’ as ‘can be found in Figure 2.’
7. The equations which do not belong to authors should be referenced. For path loss models please use the references Path loss investigation in hall environment at centimeter and millimeter-wave bands; ‘Propagation path loss prediction modelling in enclosed environments for 5G networks: a review’; ‘Path loss measurements and model analysis in an indoor corridor environment at 28 GHz and 38 GHz’.
8. Page 10: ‘in which fc is the central frequency, hBS is the base-station antenna height, hUE is the user device’s antenna height.’ should be written as ‘where fc, hBS and hUE are the central frequency, the base-station antenna height and the user device’s antenna height, respectively.’. Similarly, check and rephrase at other places.
9. The quality of results figures should be improved.
10. The axes labels for cluster formation figures should be included.
11. Please define the parameters fmax, favg and fbest.
12. In Table 2, please include the units of Pt, Pdiff, fmax, favg and fbest.
13. The text after Table 2 is the explanation about results, hence, it should be included before conclusion section.
14. Page 17 (lines 494 and 501): ‘IEEE access’ should be written as ‘IEEE Access’.
15. Page 17 (line 513): ‘Engineering optimization’ should be written as ‘Engineering Optimization’.
16. There are some typo errors in the paper, please proofread it, I would like to see the revised paper.

Experimental design

Please see the comments given in section '1. Basic reporting'.

Validity of the findings

Please see the comments given in section '1. Basic reporting'.

Additional comments

Please see the comments given in section '1. Basic reporting'.

Reviewer 2 ·

Basic reporting

The authors focus their study on the positioning problem of small cells within a 5G network by utilizing real data and they introduce two hybrid algorithms mainly based on the K means clustering approach to achieve the optimal network coverage with low power small cells. The authors have utilized real data in order to test the proposed framework and they have provided a very detailed analysis of the theoretical part of their paper as well as a detailed set of numerical results in order to show the pure operation and the performance of the proposed framework. The authors are highly encouraged to consider the following suggestions provided by the reviewer in order to improve the scientific depth of their manuscript, as well as they need to address the following minor comments in order to improve the quality of presentation of their manuscript. Initially, the provided related work needs to be substantially revised in order to be presented by using more summative language in order to better identify the research contributions that have already been proposed in the literature, as well as the research gap that the authors try to address. Furthermore, the authors need to discuss several existing approaches that have been introduced in the literature dealing with heterogeneous wireless networks, such as Unified User Association and Contract-Theoretic Resource Orchestration in NOMA Heterogeneous Wireless Networks, doi: 10.1109/OJCOMS.2020.3024778, in order to support the end users quality of service prerequisites considering a small cells development environment. Furthermore, the authors need to include a table summarizing the main notation that has been used in the paper, as well as they need to provide the units of the corresponding metrics, wherever this is appropriate. Furthermore, the authors need to include an additional section in their manuscript providing the theoretical analysis of the computational complexity of the presented algorithmic approaches and discuss if they can be implemented in real time or even close to real time manner. Furthermore, the authors need to provide some indicative numerical results capturing the real execution time of the proposed approaches and not only the iterations and discuss the real time or non real time implementation. Finally, the manuscript has several typos, syntax, and grammar errors that the authors need to address in the revised version.

Experimental design

The authors focus their study on the positioning problem of small cells within a 5G network by utilizing real data and they introduce two hybrid algorithms mainly based on the K means clustering approach to achieve the optimal network coverage with low power small cells. The authors have utilized real data in order to test the proposed framework and they have provided a very detailed analysis of the theoretical part of their paper as well as a detailed set of numerical results in order to show the pure operation and the performance of the proposed framework. The authors are highly encouraged to consider the following suggestions provided by the reviewer in order to improve the scientific depth of their manuscript, as well as they need to address the following minor comments in order to improve the quality of presentation of their manuscript. Initially, the provided related work needs to be substantially revised in order to be presented by using more summative language in order to better identify the research contributions that have already been proposed in the literature, as well as the research gap that the authors try to address. Furthermore, the authors need to discuss several existing approaches that have been introduced in the literature dealing with heterogeneous wireless networks, such as Unified User Association and Contract-Theoretic Resource Orchestration in NOMA Heterogeneous Wireless Networks, doi: 10.1109/OJCOMS.2020.3024778, in order to support the end users quality of service prerequisites considering a small cells development environment. Furthermore, the authors need to include a table summarizing the main notation that has been used in the paper, as well as they need to provide the units of the corresponding metrics, wherever this is appropriate. Furthermore, the authors need to include an additional section in their manuscript providing the theoretical analysis of the computational complexity of the presented algorithmic approaches and discuss if they can be implemented in real time or even close to real time manner. Furthermore, the authors need to provide some indicative numerical results capturing the real execution time of the proposed approaches and not only the iterations and discuss the real time or non real time implementation. Finally, the manuscript has several typos, syntax, and grammar errors that the authors need to address in the revised version.

Validity of the findings

The authors focus their study on the positioning problem of small cells within a 5G network by utilizing real data and they introduce two hybrid algorithms mainly based on the K means clustering approach to achieve the optimal network coverage with low power small cells. The authors have utilized real data in order to test the proposed framework and they have provided a very detailed analysis of the theoretical part of their paper as well as a detailed set of numerical results in order to show the pure operation and the performance of the proposed framework. The authors are highly encouraged to consider the following suggestions provided by the reviewer in order to improve the scientific depth of their manuscript, as well as they need to address the following minor comments in order to improve the quality of presentation of their manuscript. Initially, the provided related work needs to be substantially revised in order to be presented by using more summative language in order to better identify the research contributions that have already been proposed in the literature, as well as the research gap that the authors try to address. Furthermore, the authors need to discuss several existing approaches that have been introduced in the literature dealing with heterogeneous wireless networks, such as Unified User Association and Contract-Theoretic Resource Orchestration in NOMA Heterogeneous Wireless Networks, doi: 10.1109/OJCOMS.2020.3024778, in order to support the end users quality of service prerequisites considering a small cells development environment. Furthermore, the authors need to include a table summarizing the main notation that has been used in the paper, as well as they need to provide the units of the corresponding metrics, wherever this is appropriate. Furthermore, the authors need to include an additional section in their manuscript providing the theoretical analysis of the computational complexity of the presented algorithmic approaches and discuss if they can be implemented in real time or even close to real time manner. Furthermore, the authors need to provide some indicative numerical results capturing the real execution time of the proposed approaches and not only the iterations and discuss the real time or non real time implementation. Finally, the manuscript has several typos, syntax, and grammar errors that the authors need to address in the revised version.

Additional comments

The authors focus their study on the positioning problem of small cells within a 5G network by utilizing real data and they introduce two hybrid algorithms mainly based on the K means clustering approach to achieve the optimal network coverage with low power small cells. The authors have utilized real data in order to test the proposed framework and they have provided a very detailed analysis of the theoretical part of their paper as well as a detailed set of numerical results in order to show the pure operation and the performance of the proposed framework. The authors are highly encouraged to consider the following suggestions provided by the reviewer in order to improve the scientific depth of their manuscript, as well as they need to address the following minor comments in order to improve the quality of presentation of their manuscript. Initially, the provided related work needs to be substantially revised in order to be presented by using more summative language in order to better identify the research contributions that have already been proposed in the literature, as well as the research gap that the authors try to address. Furthermore, the authors need to discuss several existing approaches that have been introduced in the literature dealing with heterogeneous wireless networks, such as Unified User Association and Contract-Theoretic Resource Orchestration in NOMA Heterogeneous Wireless Networks, doi: 10.1109/OJCOMS.2020.3024778, in order to support the end users quality of service prerequisites considering a small cells development environment. Furthermore, the authors need to include a table summarizing the main notation that has been used in the paper, as well as they need to provide the units of the corresponding metrics, wherever this is appropriate. Furthermore, the authors need to include an additional section in their manuscript providing the theoretical analysis of the computational complexity of the presented algorithmic approaches and discuss if they can be implemented in real time or even close to real time manner. Furthermore, the authors need to provide some indicative numerical results capturing the real execution time of the proposed approaches and not only the iterations and discuss the real time or non real time implementation. Finally, the manuscript has several typos, syntax, and grammar errors that the authors need to address in the revised version.

---

## Round 0.2 · Minor Revisions

· Academic Editor

Minor Revisions

The citation of tables, figures, references, etc in the conclusion should be removed.

Reviewer 1 ·

Basic reporting

no comment

Experimental design

no comment

Validity of the findings

no comment

Additional comments

Authors have addressed the comments in the revised paper. The conclusion still needs improvements. Citation of tables, figures, references etc in the conclusion may be removed.

Reviewer 2 ·

Basic reporting

The authors have addressed in detail the reviewers' comments. This reviewer has no further concerns regarding this paper.

Experimental design

The authors have addressed in detail the reviewers' comments. This reviewer has no further concerns regarding this paper.

Validity of the findings

The authors have addressed in detail the reviewers' comments. This reviewer has no further concerns regarding this paper.

Additional comments

The authors have addressed in detail the reviewers' comments. This reviewer has no further concerns regarding this paper.

---

## Round 0.3 · accepted · Accept

· Academic Editor

Accept

I assessed the revised submission and the authors have addressed all the reviewer's comments and the article is ready for publication.